# Cost effectiveness and return on investment analysis for surgical care in a conflict-affected region of Sudan

C. Phifer Nicholson[1], Anthony Saxton[1], Katherine Young[2], Emily R. Smith[1,3,4], Mark G. Shrime[5,6], Jon Fielder[7], Thomas Catena[2], Henry E. Rice[1,3]*

1 Department of Surgery, Duke University School of Medicine, Durham, North Carolina, United States of America, 2 Mother of Mercy-Gidel Hospital, Gidel, Sudan, 3 Duke Global Health Institute, Duke Center for Global Surgery and Health Equity, Durham, North Carolina, United States of America, 4 Department of Emergency Medicine, Duke University School of Medicine, Durham, North Carolina, United States of America, 5 Department of Global Health and Social Medicine, Harvard Medical School, Boston, Massachusetts, United States of America, 6 Mercy Ships, Garden Valley, Texas, United States of America, 7 African Mission Healthcare, Kenya, United States of America

* henry.rice@duke.edu

**Data Availability Statement:** All data included in the supplemental information.

**Funding:** The authors received no specific funding for this work.

## Abstract

The delivery of healthcare in conflict-affected regions places tremendous strains to health systems, and the economic value of surgical care in conflict settings remains poorly understood. Our aims were to evaluate the cost-effectiveness, societal economic benefits, and return on investment (ROI) for surgical care in a conflict-affected region in Sudan. We conducted a retrospective study of surgical care from January to December 2022 at the Mother of Mercy-Gidel Hospital (MMH) in the Nuba Mountains of Sudan, a semi-autonomous region characterized by chronic and cyclical conflict. We collected data on all patients undergoing surgical procedures (n = 3016), including age, condition, and procedure. We used the MMH budget and financial statements to measure direct medical and non-medical expenditures (costs) for care. We estimated the proportion of expenditures for surgical care through a survey of surgical vs non-surgical beds. The benefits of care were calculated as averted disability-adjusted life-years (DALYa) based on predicted outcomes for the most common 81% of procedures, and then extrapolated to the overall cohort. We calculated the average cost-effectiveness ratio (CER) of care. The societal economic benefits of surgical care were modeled using a human capital approach, and we performed a ROI analysis. Uncertainty was estimated using sensitivity analysis. We found that the CER for all surgical care was $72.54/DALYa. This CER is far less than the gross domestic product per capita in the comparator economy of South Sudan ($585), qualifying it as very cost-effective by World Health Organization standards. The total societal economic impact of surgical care was $9,124,686, yielding a greater than 14:1 ROI ratio. Sensitivity analysis confirmed confidence in all output models. Surgical care in this conflict-affected region of Sudan is very cost-effective, provides substantial societal economic benefits, and a high return on investment.

**Competing interests:** The authors have declared that no competing interests exist.

## Introduction

Armed conflict causes tremendous challenges to health systems, such as interruptions in supply chains, closure of facilities, threats to effective governance, harm to medical staff, food insecurity, gender-based violence, and outbreaks of communicable diseases [1–5]. All of these challenges can be long-lasting, even after acute conflict has subsided ("chronic conflict"). Civilian populations are disproportionately impacted by armed conflict, particularly women, children, and the elderly [2]. Despite the impact of conflict on health care systems, the global health community has not provided adequate attention to understanding healthcare delivery in conflict-affected zones, including study of the economic value of care [6].

In many conflict-affected regions, there is a common perception that support of health services is resource-intensive, although empirical research on the economic value of care in these areas is limited [7]. Cost-effectiveness and return on investment (ROI) analyses take into account both the cost and benefits of care simultaneously within a structured framework, and can help policymakers assess economic factors when allocating resources [8,9]. Many areas of surgical care in LMICs have been shown to be cost-effective and offer a high ROI, although most of these studies are from health systems in secure political settings [10,11]. In contrast, the economic impact for surgical care in conflict-affected regions remains poorly understood.

Sudan has been involved in civil conflict for decades, with cyclical and chronic civil instability leading to long-term large-scale internal displacement, food insecurity, human rights abuses, conflict-related sexual violence, and attacks on medical facilities [12]. In this case study, our aims were to measure the cost-effectiveness, societal economic benefits, and ROI for surgical care at the Mother of Mercy-Gidel Hospital (MMH), located within a conflict-affected region in the Nuba Mountains region of Sudan.

## Methods

### Overview

We conducted a retrospective cohort study of all patients who underwent surgical care at MMH over a 12-month period from January to December 2022. We used the hospital budget and year-end financial statements to summarize all costs (expenditures) for care, including all direct medical costs for human resources/salaries, medications, disposables, and infrastructure as well as non-medical costs of patient and family meals. We estimated the proportion of general hospital expenditures for surgical care through a survey of beds for surgical vs non-surgical patients. The benefits of surgical care were estimated in terms of averted disability-adjusted life-years (DALYa) using predicted clinical outcomes for the most common 81% of all procedures, which were extrapolated to the overall cohort. We calculated the average cost-effectiveness ratio (CER) for all surgical procedures, with subanalysis by surgical specialty. The societal economic impact of surgical care was modeled using a human capital approach, with the ROI ratio calculated. We performed deterministic sensitivity analysis to estimate the uncertainty of all input variables on all model outputs.

### Setting

Sudan has been in a state of recurring and chronic armed conflict since the country gained independence in 1956. The Nuba Mountains within the South Kordofan state is part of the "Three Areas" region of Sudan which has been characterized by high levels of cyclical armed conflict for the past several decades and has been designated by the international community for special concern [13]. The region has been excluded from most types of support from the national government, with a limited semi-autonomous government overseen by the local rebel

organization (Sudan People's Liberation Movement-North). The chronic conflict in the Nuba Mountains has led to long-term collapse of local health systems, massive internal displacement, food insecurity, loss of government infrastructure, large-scale human rights abuses, and violations of international law, including conflict-related sexual violence and attacks on medical facilities [12,14].

The Mother of Mercy-Gidel Hospital is a 425-bed hospital within the Nuba Mountains of Sudan. The region and hospital itself has been bombed several times over the past few decades by the Sudanese Air Force [15]. MMH was formed in 2008 in partnership with the regional government and the Catholic Diocese of El Obeid, Sudan. The hospital is one of two sources of surgical care in the region, with one other hospital providing basic surgical care, including Cesarean section. The hospital has a geographic catchment area similar in size to the country of Austria. The catchment area has approximately 2,000,000 people, although there are large populations of internally displaced persons, making exact population estimates difficult [14]. The hospital supports a range of inpatient and outpatient services, including medical and pediatric care, maternal health, mental health, as well as treatment for tuberculosis and leprosy.

The hospital provides surgical care in general surgery, obstetrics and gynecology, orthopedics, ophthalmology, urology, neurosurgery, and dental care, among other areas. Full-time medical staff includes a family physician trained in multiple surgical disciplines who performs most of the surgical operations as well as two other physicians. The hospital is supported by short-term visits by international clinicians who work with the full-time staff, although 93% of the surgical procedures in 2022 were performed by MMH full-time staff as the primary surgeon. A mid-level provider trained in ophthalmological care performs cataract removals and other ophthalmologic procedures, and a mid-level provider supports dental care. Anesthesia care is provided by three nurse anesthetists. The hospital performs the three bellwether procedures suggestive of a high-functioning surgical system as defined by the Lancet Commission on Global Surgery, including Cesarean section, laparotomy, and treatment of open fracture [16].

MMH has a single overall budget for all operations, with administration supported by African Mission Healthcare. Financial support at MMH is through contributions from donors (>97% of revenues), with smaller contributions from non-government organizations and patient fees (1.4% of revenues). A single overall budget is used to track all hospital expenditures regardless of contributing source. Patients are requested to offset costs with out-of-pocket expenditures depending on type of care. There is a flat rate for all consultations (approximately $1), a flat weekly rate for hospitalizations (approximately $20), and three different levels of payment for surgical procedures of approximately $5, $10, or $50 depending on the type of procedure (minor, major, or complex). No patient is refused care for inability to pay, and approximately 50–60% of patients do not pay any out-of-pocket expenditures.

### Data collection

We conducted a retrospective review of the surgical logbooks of all patients undergoing inpatient or outpatient surgical procedures over a 12-month period from January 2022 to December 2022, with the data themselves collected from January to June 2023. Data were extracted from the logbooks, entered into an Excel spreadsheet with all protected health information removed, and stored on a secure, encrypted computer. We collected data on patient age, gender, diagnosis, and primary surgical procedure performed. Other data such as patient comorbidities, complications, or clinical outcomes were not collected.

Overall hospital budget expenses in 2022 were obtained from the MMH and African Mission Healthcare general ledgers and MMH year-end financial statements. As detailed above,

this amount includes expenditures for all direct medical and non-medical costs as well as administrative costs. Final surgical expenses included the entire amount of expenses directly related to care of surgical patients as well as the portion of general health system expenditures attributable to surgical care.

## Cost of surgical care

The costs of surgical care were estimated from a general societal view [17]. We used the hospital annual budget to collect all expenses from 2022, with an additional 15.82% based on estimated administrative overhead support from African Mission Healthcare using their standard practices. To calculate the proportion of expenditures for surgical care, we first identified budget items directed only for surgical care (operating room staff salaries, surgical programs) and attributed these costs to surgical expenses. The remaining general expenditures included support for all direct medical costs such as staff salaries/benefits, supplies, administrative overhead, maintenance, medications, disposables, as well as direct non-medical costs for patient and family meals. As these costs are shared across all service lines, we assumed the percentage of the general expenditures allocated to surgical care was equivalent to the proportion of inpatient hospital beds occupied by surgical patients.

To determine the proportion of beds attributable to surgical care, we conducted a daily survey of hospital beds over a five-day period in January 2022, with one researcher assigning each bed to surgical vs non-surgical based on the health needs of that patient. We chose a five-day period based on convenience sampling, although we compared our results with several other time periods to account for seasonal variation in bed allocation. Patients were defined as surgical if they underwent any surgical procedure during that hospital admission or were hospitalized for other surgical needs (e.g., burn dressings, wound infection, etc.). Finally, we included expenses for depreciation of the hospital building and equipment as reported in the MMH financial statements for 2022, and attributed the portion of depreciation for surgical care based on the proportion of hospital beds for surgical patients. For analysis of cost per case, we assumed the same costs for all procedures.

The presence of a visiting clinician did not substantially impact the ratio of surgical vs non-surgical beds. For example, for the 5-day period of analysis, only 2–3 of the hospital beds each day were directed towards the surgical procedure of a visiting surgeon. However, there is some variation in bed allocation due to seasonal changes in malaria prevalence. Malaria is the highest single admission diagnosis and in 2022 accounted for 13.6% of all inpatient admissions. Malaria only accounted for 2.7% of admissions in March. In contrast, malaria accounted for 36.1% of admissions in October. We accounted for variation in surgical vs non-surgical bed allocation in our sensitivity analysis (see below).

## Benefits of surgical care

The benefits of surgical care were estimated by calculating the averted DALYs (DALYa) based on predicted clinical outcomes for each surgical procedure based on established clinical studies. The DALYa represents the amount of premature death or disability attributable to a surgical condition that is avoided through a surgical intervention. The DALY is a widely used metric of disease burden, initially described in the Global Burden of Disease (GBD) studies [18].

We estimated the DALYa for each individual surgical procedure for the most commonly performed procedures (procedures performed more than 10 times) and then extrapolated to the overall cohort of patients using established methods for extrapolation of economic health data [19,20]. We chose to use this extrapolation process, as it was not practical to determine all

of the required input variables for the many surgical procedures which were performed extremely rarely. We assumed that the extrapolated cohort of procedures had similar benefits (DALYa) to the selected sample.

We used standard nomenclature for DALY calculations (r, K, β), which specifies the discount rate (r), age-weighting modulation (K), and age-weighting parameter (β) factored into DALY calculations [11,21]. The baseline averted DALY estimates were calculated using a 3% discount rate with no age-weighting (3, 0, 0). Averted DALY calculations were performed using the equation of Shrime et al. [21], taking into account the expected outcome of surgical procedure, risk of complications, and expected outcome with complications:

$$DALYa = YLL(RD - RD_{postTx}) + PST(RD_{postTx} \times YLL + YLD_{dz} - pCompl \times YLD_{compl})$$

where:

$YLL$ = years of life lost,

$RD$ = risk of death with no treatment,

$RD_{postTx}$ = risk of death following treatment,

$PST$ = probability of successful treatment,

$YLD_{dz}$ = years lived with disability following an unsuccessful treatment,

$pCompl$ = probability of complications arising after a successful treatment,

$YLD_{compl}$ = years lived with disability due to complications after successful treatment.

For each patient, the years of life lost (YLL) were calculated using the formula [22]:

$$YLL = \frac{KCe^{ra}}{(r+\beta)^2}\left[e^{-(r+\beta)(L+a)}[-(r+\beta)(L+a)-1] - e^{-(r+\beta)a}[-(r+\beta)a-1]\right]$$
$$+ \frac{1-K}{r}(1-e^{-rL})$$

where variables represent:

$K$ = age-weighting modulation constant (0 for no age-weighting, 1 for age-weighting),

$C$ = the adjustment constant for age-weights (0.1658),

$e$ = natural logarithm root (2.72),

$r$ = discount rate,

$a$ = predicted age of death without treatment

$\beta$ = age weighting constant (0.04),

$L$ = standard life expectancy in South Sudan at age of death (by sex and age) [23].

The years lived with disability (YLD) were calculated using the formula:

$$YLD = DW\left\{\frac{KCe^{ra}}{(r+\beta)^2}[e^{-(r+\beta)(L+a)}[-(r+\beta)(L+a)-1] - e^{-(r+\beta)a}[-(r+\beta)a-1]] + \frac{1-K}{r}(1-e^{-rL})\right\}c$$

where variables represent:

$DW$ = disability weight,

$L$ = duration of disability,

$K, C, e, r, a, \beta$ = same as above.

We calculated the average cost-effectiveness ratio (CER) for all surgical care as well as stratified by surgical type by the formula [24]:

$$CER = \text{aggregate surgical costs/aggregate DALYa}$$

## Data assumptions and sources

We used a range of clinical studies and datasets to inform all input variables (**Tables A, B, and C in S1 Appendix**). We note that most parameter assumptions (such as PST, RD, etc.) were informed by clinical studies mainly from high income settings, as there are very little available data on these parameters from conflict settings or LMICs.

We assigned disability weights based on the 2019 Global Burden of Disease (GBD) study [18]. For conditions that did not have assigned DWs, we used a DW from a similar condition. The years lived with disability due to complications after successful treatment ($YLD_{compl}$) was accounted for using similar DWs, with the probability of complications arising after a successful treatment (pCompl) based on existing studies. Finally, we assigned each procedure to a different type of surgical specialty and summarized the proportion of cases for each surgical specialty. We determined the average CER for each procedure and summarized CER for each surgical specialty.

## Societal economic impact and return on investment

We modeled the societal economic impact of surgical care using a human capital approach. This approach estimates the societal economic losses that are avoided as a result of surgical treatment, and equates the value of years of human life to the market value of the economic output produced by an individual over a lifetime [25]. To calculate the societal economic impact of surgical treatment, the DALYa for each patient were multiplied by purchasing power parity estimates of gross national income (GNI) per capita in South Sudan from 2015 inflated to current United States dollars, which was $1040 [26].

The social ROI is a framework that helps to quantify the social, environmental and/or economic value being created by producing a ratio that states how much social value (in monetary terms) is created for every unit of investment [8]. We calculated the ROI for surgical care in Sudan by the formula:

$$\text{Return on investment} = \text{overall societal economic benefits}/\text{overall costs}$$

We used South Sudan as a national comparator for economic analysis, as the local economy and political systems in the Nuba region are more closely aligned to South Sudan than to Sudan [27]. After the partition of Sudan and South Sudan in 2011, the Nuba Mountains remained in Sudan despite siding with the South Sudan region politically during the civil war. Since 2011, the Nuba Mountain region has largely remained economically and politically cut off from the rest of Sudan [13,27]. Many Nuba refugees have fled to South Sudan over this time period, and the majority of supply chain items are transported to MMH through South Sudan and surrounding countries.

## Sensitivity analysis

We used one-way and two-way deterministic sensitivity analyses to test the impact of uncertainty of all input variables on the upper and lower bounds of the cost-effectiveness and societal benefit model outputs [28]. We calculated the upper and lower bound values of both outputs with no discounting or age-weighting (0, 0, 0), values with a 3% discount weight with 4% age-weighting (3, 1, 0.04), as well as with uncertainty in the extrapolation of costed data, allocation of surgical vs non-surgical beds, risk of death, probability of successful treatment, predicted age of death without treatment, disability weight, duration of disability, property value, equipment value, probability of complications, and disability weight with complications.

For one-way sensitivity analysis, we multiplied individual baseline input variables ±15% (with a maximum of 100% where applicable) or changed discounting and age-weighting (yes/no) while all other input variables remained unchanged. Results of one-way sensitivity analyses were summarized in tornado diagrams, showing the impact of the uncertainty of each input variable on the upper and lower bounds of model outputs. For two-way sensitivity analyses, we calculated the impact of varying all variables in similar fashion simultaneously on the upper and lower bounds of model outputs.

## Statistical analysis

We summarized all data using Excel (Microsoft Corp., Redmond, WA) and exported it to STATA v14.2 (StataCorp, College Station, TX) for analysis. Results for costs, averted DALYs, and economic impact were calculated as median values with the interquartile range (IQR) of all surgical care and stratified by surgical specialty.

## Ethics statement

The study was approved by the Duke University Institutional Review Board (approval number 2023–0205). Although there is not a local IRB in the Nuba Mountains region of Sudan available for ethics review, both the MMH hospital administrator and the Secretariat of Health in the Nuba Mountains (run by the local rebel government) conducted an ethical review and approved this study prior to the collection of any data. As this observational, non-interventional study collected only deidentified patient data, no informed consent from individual patients was obtained. Note that this research was conducted in alignment with the Consolidated Health Economic Evaluation Reporting Standards 2022 (CHEERS 2022) statement.

## Patient and public involvement

Our study design was conceived by a multidisciplinary group of researchers at the Mother of Mercy Hospital in Sudan, African Mission Healthcare in Kenya, and the Duke Global Health Institute in the USA. This study was designed to represent the varied interests of patients, clinicians, policy makers, non-government partners, and governmental agencies.

Although due to the challenging aspects of healthcare delivery in conflict settings it was not possible to directly involve patients in study design or conduct, the concept of patient involvement translated to the study design to ensure that our outcomes were framed to optimize surgical care for patients in conflict settings. As results emerge, we will ensure that we present the findings in the most effective way beyond the research community to the general population.

## Results

### Demographics and types of surgical procedures

A total of 3016 total surgical procedures were conducted at MMH in 2022 and formed the overall cohort for this study (**Table 1**). The average age of the patients was 36 years, with a range of two days to 99 years. There were relatively equal numbers of males and females. The most commonly performed procedures (>10 times each, n = 2435 procedures total) represented 81% of all procedures. This sample was used for DALYa calculations (**Table 2**), with the benefits of surgical care extrapolated to the overall cohort. The most common types of surgical procedures were general surgery, ophthalmology, and obstetrics and gynecology.

**Table 1. Demographics of patients undergoing surgical care at Mother of Mercy Hospital in Sudan (N = 3016).**

| Age (yrs) | Number (%) |
|---|---|
| <1 | 75 (2%) |
| 1–5 | 178 (6%) |
| 6–10 | 138 (5%) |
| 11–18 | 652 (22%) |
| 19–30 | 351 (12%) |
| 31–40 | 403 (13%) |
| 41–50 | 286 (9%) |
| 51–60 | 281 (9%) |
| 61–70 | 349 (12%) |
| 71–80 | 239 (8%) |
| >81 | 64 (2%) |
| **Gender** | **Number (%)** |
| Female | 1538 (51%) |
| Male | 1478 (49%) |

Demographics of and types of procedures for all patients undergoing surgical care at Mother of Mercy Hospital in Sudan in 2022.

## Costs of surgical care

The overall annual expenses at MMH in 2022 were $2,026,366. Of that amount, $138,866 was spent on depreciable assets, and $1,887,500 was spent to operate the hospital and clinics. Of the $1,887,500, $48,180 was directly attributable to surgical expenses, and the remaining $1,839,320 was for the overall care of all surgical and non-surgical patients.

Results of the bed survey found that 30.9% of the inpatient hospital beds were for surgical patients (**Table 3**). Assuming that the proportion of surgical beds in the hospital reflects the proportion of the remaining general expenditures.

## Benefits of surgical care

The benefits of surgical care were calculated as averted DALYs for the most common 81% of procedures, and then extrapolated to the overall cohort. The benefits of all surgical care at MMH over the one-year period was 8,774 averted DALYs, with maternal health, general surgery, and orthopedic surgery accounting for the most DALYs.

The average CER for all surgical care at MMH was $72.54/DALYa. This CER is far less than the GDP per capita in South Sudan in 2022 ($585) [29], qualifying it as very cost-effective by World Health Organization (WHO) standards [30]. We found wide variability in CERs by surgical specialty, with obstetrics and gynecology ($15.70/DALYa), plastic surgery ($19.77/DALYa), and orthopedic surgery ($21.41/DALYa) the most cost-effective specialties (lowest CERs), and dental care the least cost-effective specialty ($5,495.67/DALYa) (**Table D in S1 Appendix**).

## Societal impact of surgical care and return on investment

The total societal economic benefit of surgical care using a human capital approach for the most common 81% of procedures was $7,366,913, which extrapolates to total benefits of $9,124,686 for all surgical care (**Table 4**). These economic benefits lead to over a 14:1 ROI ratio for each dollar invested in surgical care.

**Table 2. Surgical conditions treated and surgical specialty at the Mother of Mercy Hospital in Sudan.**

| Condition Treated | Surgical Specialty | Number (% Cases) |
| --- | --- | --- |
| Cataract | Ophthalmology | 610 (25%) |
| Abscess | General surgery | 335 (14%) |
| Dental | Oral and maxillofacial surgery | 229 (9%) |
| Hernia-related | General surgery | 139 (6%) |
| Miscarriage / blighted ovum | Obstetrics and Gynecology | 122 (5%) |
| Obstructed labor | Obstetrics and Gynecology | 119 (5%) |
| Necrotic wound / gangrene | General surgery | 98 (4%) |
| Foreign body | General surgery | 90 (4%) |
| Laceration | General surgery | 76 (3%) |
| Foreign body, eye | Ophthalmology | 69 (3%) |
| Hydrocele | General surgery | 44 (2%) |
| Fracture / Non-union | Orthopedic surgery | 43 (2%) |
| Lipoma | General surgery | 43 (2%) |
| Burn | Plastic surgery | 37 (2%) |
| Benign prostatic hypertrophy | Urology | 28 (1%) |
| Urologic stricture | Urology | 27 (1%) |
| Ear pain/wax/pus | General surgery | 25 (1%) |
| Chronic osteomyelitis | Orthopedic surgery | 25 (1%) |
| Anal fissure | General surgery | 24 (1%) |
| Implant | Plastic surgery | 18 (1%) |
| Ectopic pregnancy | Obstetrics and Gynecology | 17 (1%) |
| Subcutaneous mass, various | General surgery | 17 (1%) |
| Urologic stone | Urology | 17 (1%) |
| Thyroid goiter | General surgery | 16 (1%) |
| Breast cancer/fibroadenoma | General surgery | 14 (1%) |
| Hydrocephalus (congenital) | Neurosurgery | 14 (1%) |
| Unknown mass/lymphadenopathy | General surgery | 14 (1%) |
| Appendicitis | General surgery | 13 (1%) |
| Perianal fistula | General surgery | 13 (1%) |
| Circumcision | General surgery | 12 (<1%) |
| Colorectal/anorectal cancer | General surgery | 11 (<1%) |
| Extremity amputation | Orthopedic surgery | 11 (<1%) |
| Fibroid uterus | Obstetrics and Gynecology | 11 (<1%) |
| Prostate cancer | Urology | 11 (<1%) |
| Small bowel obstruction | General surgery | 10 (<1%) |
| Urinary retention | Urology | 10 (<1%) |
| Vesicovaginal fistula | Obstetrics and Gynecology | 10 (<1%) |
| Uterine rupture | Obstetrics and Gynecology | 8 (<1%) |
| Eclampsia | Obstetrics and Gynecology | 5 (<1%) |

Types of surgical cases and distribution across surgical specialities for the most common 81% of cases included in costing analysis (n = 2435).

## Sensitivity analysis

Sensitivity analysis showed relative confidence in both the CER and societal economic benefits models. By one-way sensitivity analysis, the use of discounting and uncertainty in extrapolation of costed cases had the greatest impact on the upper and lower bounds of both the CERs (**Table E in S1 Appendix; Fig 1A**) and societal economic models (**Table F in S1 Appendix; Fig 1B**).

Table 3. Survey of surgical vs non-surgical beds at the Mother of Mercy Hospital in Sudan.

| Day | Surgical | Non-Surgical | Total Beds | Proportion of Surgical Beds (%) |
|---|---|---|---|---|
| 1 | 114 | 270 | 384 | 29.7% |
| 2 | 125 | 262 | 387 | 32.3% |
| 3 | 112 | 270 | 382 | 29.3% |
| 4 | 137 | 267 | 404 | 33.9% |
| 5 | 103 | 253 | 356 | 28.9% |
| Average | 118 | 264 | 383 | 30.9% |

Proportion of hospital in-patient beds for surgical vs non-surgical patients at the Mother of Mercy Hospital in Sudan. Survey conducted daily for five consecutive days with mean proportion of surgical beds summarized.

Two-way sensitivity analysis showed the lower and upper bounds of the CER (**Table G in S1 Appendix**) and societal economic impact (**Table H in S1 Appendix**) models by varying all input variables simultaneously. As compared to the baseline overall CER of $72.54/DALYa, the lower bound of the CER was $25.36/DALYa and the upper bound was $143.67/DALYa. Even with a worst-case scenario with the highest level of uncertainty of all input variables, the upper bound of the CER is still far lower than the GDP/capita in South Sudan in 2022 ($585), confirming that surgical care at MMH is very cost-effective by WHO criteria. Two-way sensitivity analysis of the societal economic impact model showed a lower bound of $5,271,246 and an upper bound of $22,378,488.

## Discussion

We found large economic benefits to surgical care in a conflict-affected zone in Sudan. Several outcomes were used to assess the economic benefits of surgical care, including analysis of the cost-effectiveness analysis, societal economic benefits, and return on investment. In our study, surgical care at MMH was very cost-effective by WHO standards, and is comparable to common public health interventions such as antiretroviral therapy for HIV and bed netting for malaria prevention [31,32]. There were large societal economic benefits, yielding over a 14:1 ROI ratio for surgical care. Given how fragile global economies are within conflict zones, even

Table 4. Total societal economic impact of surgical care at Mother of Mercy Hospital.

| Category | Cases | Economic Impact* |
|---|---|---|
| General surgery | 994 | $1,252,787 |
| Neurosurgery | 14 | $75,721 |
| Obstetrics and Gynecology | 292 | $4,082,313 |
| Ophthalmology | 679 | $221,055 |
| Oral and maxillofacial surgery | 229 | $9,145 |
| Orthopedic surgery | 79 | $809,756 |
| Plastic surgery | 55 | $610,633 |
| Urology | 93 | $305,503 |
| **Total** | **2,435** | **$7,366,913** |
| **Total after extrapolation** | **3,016** | **$9,124,686** |

*Note that societal economic impact was measured using a Human Capital approach considering averted disability-adjusted life years (DALYs) with 3% discounting and no age weighting (3,0).

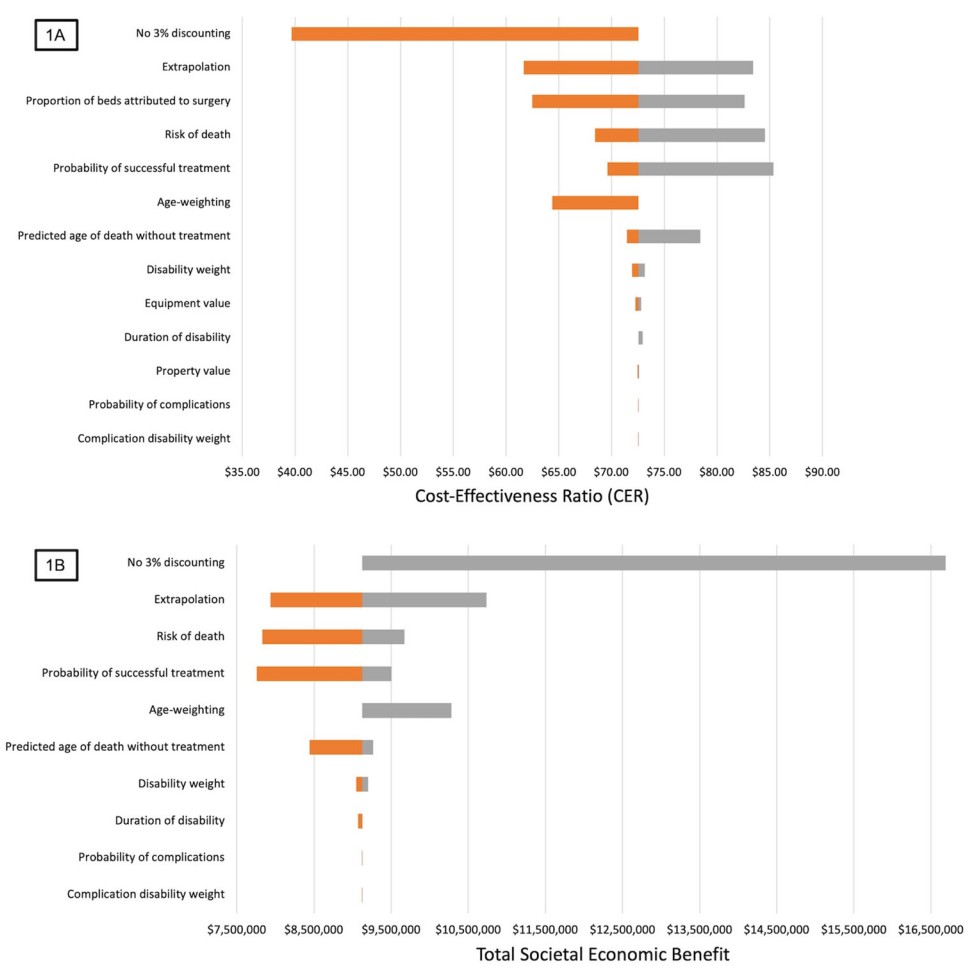

**Fig 1.** One-way sensitivity analysis for the average cost-effectiveness ratio (CER) (**Fig 1A**) and societal economic benefit (**Fig 1B**) models. Tested input variables include use of discounting (yes/no), as well as uncertainty in extrapolation of costed data to overall cohort, proportion of beds attributable to surgery, risk of death, probability of successful treatment, discounting, predicted age of death without treatment, age weighting, disability weight, duration of disability, property value, equipment value, probability of complications, and disability weight with complications. Lower bounds (orange) and upper bounds (grey) depict the impact of the uncertainty of each input variable on model outputs.

modest investments to support surgical care can markedly benefit the health and economy of local communities.

Civilians often lack basic medical care in conflict regions. Both acute and long-term conflict can lead to massive internal population displacement, which exacerbates existing health disparities among affected populations in already strained health systems [33]. Despite the challenges of healthcare delivery in conflict settings, surgical care can be provided in a high-quality and safe manner to civilian populations, as has been shown in several recent contexts [34,35]. In our study, we confirmed that surgical care can be delivered to a vulnerable population within a conflict-affected setting in Sudan in an extremely cost-effective fashion, yielding substantial societal economic benefits as well as high return on investment.

Equitable access to surgical care supports long-term economic development and productivity, particularly in low- and middle-income countries (LMICs) [16]. At a macroeconomic level, surgical care in LMICs can provides large national economic benefits such as increase in

gross domestic product [16]. At a microeconomic level, surgical care can protect families from falling into poverty due to catastrophic out-of-pocket expenditures [36]. Although the economic benefits of surgical care in LMICs are relatively well defined within stable political settings, the economic impact of surgical care in conflict zones in both LMICs as well as high-income countries has not been closely examined to date. Our study confirms that at MMH within a conflict-affected region of Sudan, surgical care is very cost effective and offers high levels of economic benefits to local economies.

Most economic analyses in global surgery use a micro-costing approach by identifying the units and costs for all fixed and variable inputs to provide surgical care, which can be difficult to measure, particularly in conflict settings [37]. The budget at MMH is relatively unique in that a single funding stream makes tracking overall hospital allocations relatively straightforward and facilitates a top-down economic analysis. The extremely low-cost structure of MMH differentiates it from many other hospitals not just in high-income settings but also in many LMICs. By context, the _annual_ operating budget at MMH is less than the _daily_ budget for similar size health systems in the U.S [38].

We do not suggest that provision of surgical care in conflict-affected regions should be based entirely on the economic value of care. Humanitarian action itself is a fundamental moral activity grounded in the ethical basis of provision of assistance to those in need as well as support of the basic rights to health [39]. Health care for civilians in conflict settings is based on core principles of humanity, neutrality, impartiality, and independence, and is deeply related to established frameworks of social justice and codified within international human rights law such as in the Geneva Conventions of 1949 [40]. However, the economic impact of healthcare in conflict-affected regions has seldom been considered in the evaluation of global health interventions, and there is an increasing interest among donors in maximizing return on investments [8,41]. Our findings suggest that surgical care is this conflict setting is not only morally just, but is also very cost-effective and may offer substantial economic benefits to help rebuild fragile economies. Importantly, our findings show a 14:1 ROI ratio in this conflict-affected region, which is even greater than the often cited 10:1 ROI ratio for surgery coverage globally from the Lancet Commission on Global Surgery [16].

## Limitations

Our study has several limitations. Most importantly, as with any economic modeling, the validity of the results is dependent on choices in the underlying assumptions, accuracy, and uncertainty of the input variables. For example, we coded every Cesarean section for obstructed labor and every amputation as related to diabetes due to limited data on other diagnoses, raising concern of overestimation of the cost-effectiveness of this care. In contrast, we coded all dental procedures as mild dental health conditions, likely far underestimating the cost-effectiveness of dental care. As well, we used predicted clinical outcomes rather than primary collected clinical outcome data, as primary data was not available. We also chose to not use age weighting (although we included age weighting in our sensitivity analysis), as the use of age weighting continues to be debated in the health economics literature. Despite all of these challenges, large number of assumptions, and uncertainty of input parameters, our sensitivity analysis showed relatively high confidence in the CER and societal economic impact models. Even with a worst-case scenario of the highest levels of uncertainty of all input variables, the surgical care at MMH is still very cost-effective by WHO criteria by a wide margin and has a high ROI ratio.

There are a number of other limitations. First, DALYs are designed to measure the burden of health conditions at a population level, not at an individual level, and accurate estimation of

the economic benefits from individual surgical procedures is challenging. Second, we used a survey of surgical vs non-surgical beds over a short time period to estimate expenditures for surgical care, although our sensitivity analysis showed relative confidence in our analysis when accounting for seasonal bed variation. Third, the direct non-medical costs of travel, in-kind contributions from visiting clinicians, and indirect opportunity costs of lost time were not included. Fourth, we estimated the benefits of surgical care as averted DALYs only from surgical procedures, which does not account for the additional benefits of non-procedural surgical care (e.g., burn dressings, etc.). However, as our CER calculations included expenditures but did not incorporate any benefits of non-procedural care, this process actually leads to underestimation of the cost-effectiveness of surgical care. Fifth, we assumed that costs were equivalent across all procedures to facilitate our analysis. Although this led to overestimation as well as underestimation of the CER for different surgical specialties, our sensitivity analyses found it did not affect our overall findings. Sixth, we compared surgical care to no treatment rather than to alternative treatments (when available), which may falsely inflate the cost-effectiveness of surgical care. However, the wide variability and lack of data on outcomes of alternative treatments make such comparisons impossible. Finally, our data was from a single hospital with an exceedingly low-cost structure located within a semi-autonomous region of Sudan, and these findings may not be generalizable to other acute or chronic conflict-affected settings.

## Conclusion

In conclusion, we found that surgical care provided at MMH within a conflict-affected zone of Sudan is very cost-effective, provides substantial societal economic benefits, and has a high return on investment. Although our case study was conducted in a single setting, it does suggest several policy recommendations:

- • If regions of both acute as well as chronic civil conflict, health care planning should include strategic support for surgical services.

- • Modest investments in surgical care offer tremendous value for money to address key health needs in conflict regions.

- • Expansion of surgical capacity, human resources, and infrastructure in the Nuba Mountains region of Sudan may provide a cost-effective approach to address local health needs and improve the regional economy in this fragile setting.

## Supporting information

**S1 Checklist. CHEERS 2022 checklist.**
(DOCX)

**S1 Appendix.** Table A. Disability Weight Assignments. Table B. Probability of Successful Treatment Assignments. Table C. Risk of Death Assignments. Table D. Cost-Effectiveness Ratios Across Surgical Specialties. Table E. One Way Deterministic Sensitivity Analysis for Cost-Effectiveness Ratios for Surgical Care at Mother of Mercy Hospital. Table F. One Way Deterministic Sensitivity Analysis for Societal Economic Impact of Surgical Care at Mother of Mercy Hospital. Table G. Two Way Deterministic Sensitivity Analysis for Cost-Effectiveness of Surgical Care at Mother of Mercy Hospital. Table H. Two Way Deterministic Sensitivity Analysis for Societal Economic Impact of Surgical Care at Mother of Mercy Hospital. Table I. References for Appendices Tables. Table J. Author Reflexivity Statement.
(DOCX)

## Author Contributions

**Conceptualization:** C. Phifer Nicholson, Emily R. Smith, Jon Fielder, Henry E. Rice.

**Data curation:** C. Phifer Nicholson, Anthony Saxton, Katherine Young, Emily R. Smith, Jon Fielder, Thomas Catena, Henry E. Rice.

**Formal analysis:** C. Phifer Nicholson, Anthony Saxton, Mark G. Shrime, Jon Fielder, Henry E. Rice.

**Investigation:** C. Phifer Nicholson, Katherine Young, Emily R. Smith, Henry E. Rice.

**Methodology:** C. Phifer Nicholson, Jon Fielder, Thomas Catena, Henry E. Rice.

**Project administration:** Henry E. Rice.

**Supervision:** Henry E. Rice.

**Validation:** Anthony Saxton, Henry E. Rice.

**Visualization:** Anthony Saxton, Henry E. Rice.

**Writing – original draft:** C. Phifer Nicholson, Anthony Saxton, Henry E. Rice.

**Writing – review & editing:** C. Phifer Nicholson, Anthony Saxton, Katherine Young, Emily R. Smith, Mark G. Shrime, Jon Fielder, Thomas Catena, Henry E. Rice.

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
