## [Decision Letter · Decision Letter 0]

21 Aug 2024

PGPH-D-24-01395

Cost Effectiveness and Return on Investment Analysis for Surgical Care in a Conflict Zone of Sudan

Dear Dr. Rice,

Thank you for submitting your manuscript to PLOS Global Public Health. After careful consideration, we feel that it has merit but does not fully meet PLOS Global Public Health’s publication criteria as it currently stands. Therefore, we invite you to submit a revised version of the manuscript that addresses the points raised during the review process.

We look forward to receiving your revised manuscript.

Kind regards,

Dhananjaya Sharma, MS, PhD, DSc, FRCS

Academic Editor

Journal Requirements:

1. Please include a complete copy of PLOS’ questionnaire on inclusivity in global research in your revised manuscript. Our policy for research in this area aims to improve transparency in the reporting of research performed outside of researchers’ own country or community. The policy applies to researchers who have travelled to a different country to conduct research, research with Indigenous populations or their lands, and research on cultural artefacts. The questionnaire can also be requested at the journal’s discretion for any other submissions, even if these conditions are not met.  Please find more information on the policy and a link to download a blank copy of the questionnaire here: https://journals.plos.org/plosone/s/best-practices-in-research-reporting. Please upload a completed version of your questionnaire as Supporting Information when you resubmit your manuscript.

Additional Editor Comments (if provided):

Needs major revision in view of strong criticism from one of the reviewers and other comments by 2nd reviewer

Reviewers' comments:

Reviewer's Responses to Questions

**Comments to the Author**

1. Does this manuscript meet PLOS Global Public Health’s publication criteria? Is the manuscript technically sound, and do the data support the conclusions? The manuscript must describe methodologically and ethically rigorous research with conclusions that are appropriately drawn based on the data presented.

Reviewer #1: No

Reviewer #2: Yes

2. Has the statistical analysis been performed appropriately and rigorously?

Reviewer #1: Yes

Reviewer #2: Yes

3. Have the authors made all data underlying the findings in their manuscript fully available (please refer to the Data Availability Statement at the start of the manuscript PDF file)?

Reviewer #1: Yes

Reviewer #2: Yes

4. Is the manuscript presented in an intelligible fashion and written in standard English?

Reviewer #1: No

Reviewer #2: Yes

5. Review Comments to the Author

Reviewer #1: This paper assessed the socioeconomic cost-effectiveness of surgical care in the Nuba Mountains region of Sudan. The authors have performed a statistically rigorous analysis. However, I have several major concerns:

1. Sudan is presented as a “conflict setting” throughout the manuscript. However, this study uses data from 2022, under a ceasefire agreement. The authors do not mention the state the study takes place in, and mostly refer to the setting as “Sudan.” The data presented in this study is not from an active conflict setting. Conflict-affected may be a more appropriate terminology, although a clear description of what is meant would be required. Overall, a lack of clarity and even misleading description of the context is concerning.

2. Sudan was not in a civil war during 2022. Conflation with the current civil war throughout the paper is misleading.

3. The authors suggest local ethical oversight was not possible due to the ongoing conflict. While the data was generated during a ceasefire, data extraction took place during the current civil war (throughout 2023). However, it is not acceptable to have no local ethical oversight. Presumably, if local data extraction was possible, local ethical oversight would be as well. Research in active conflict settings have relied on community advisory boards. Other studies in Sudan have received ethical oversight from the General Directorate of Research of the Federal Ministry of Health.

4. Armed conflict “causes tremendous challenges to health systems” in countries of all economic classifications.

5. While the Discussion section mentions humanitarian reasons to provide surgical care, there is no mention of the right to health. It is unclear what international humanitarian law the authors reference (which is different that international human rights law).

Reviewer #2: Little research has been conducted in conflict areas due to instability and logistical challenges conducting research in these settings. Research on these settings is limited, and there is an existing perception that surgery is too expensive in resource-limited settings. The authors should be applauded for their efforts and for shedding light on the cost-effectiveness of surgical services provision in resource-limited conflict areas.

The authors conducted a retrospective cohort study of 3,016 patients presenting over a 12 month period in a representative catchment hospital (Mother of Mercy-Gidel Hospital, a 425-bed hospital which is the only source of tertiary surgical care in the region). Staff is limited to a family physician trained in multiple surgical disciplines performing most of the surgical operations and two other physicians, mid-level ophthalmological provider, and mid-level dental provider, and three nurse anesthetists.

A single overall budget is used to track all hospital expenditures. This comprehensively assessed cost, including all direct medical costs for human resources/salaries, medications, disposables, and infrastructure as well as non-medical costs of patient and family meals. However, a micro-costing approach was not used, and for analysis of cost per case, they assumed the same costs for all procedures. Cost extrapolation was also based on commonly performed procedures (defined as >10 instances over the year), which was reasonable. Their sample of procedures each performed at least 10 times over the year represented 81% of all cases. Without clinical outcomes data, DALYs were modeled on predicted clinical outcomes.

The overall cost per surgical case was $211, average CER for all surgical care at MMH was $72.54/DALYa (less than Sudanese GDP/capita), determining CER is very cost-effective, further confirmed with appropriate sensitivity analyses. ROI was projected to be 14:1 for each dollar invested in surgical care in the study setting. Variability in CERs across surgical specialties was very interesting, with obstetrics and gynecology 367 ($15.70/DALYa), plastic surgery ($19.77/DALYa), and orthopedic surgery 368 ($21.41/DALYa) the most cost-effective specialties.

This was a very interesting study, shedding light on the provision of surgical care in a conflict area. Their conclusion that surgical care is conflict settings is not only morally just, but is also cost-effective and offer substantial economic benefits to help rebuild fragile economies is reasonable. The study design was robust, but inputs were inherently weaken without clinical outcomes data and relied heavily on informed estimates. However, sensitivity analyses were reassuring, and the authors correctly acknowledge prospectively assigning cost on a per patient basis would’ve made this a stronger study, using a micro-costing approach.

I would like the authors to further address:

what percentage of patients participate in the program in which patients are requested to offset costs with out-of-pocket expenditures from $1-$50 depending on type of care, although no patient is refused care for inability to pay?

Additionally, the authors mention flat rates. What are these rates specifically? (“a flat rate for all consultations, a flat weekly rate for hospitalizations, and three different levels of payment for surgical procedures (minor, major, or complex)”).

Page 7 line 202 - missing verb “use/utilize” between “chose to” and “this extrapolation”

6. PLOS authors have the option to publish the peer review history of their article (what does this mean?). If published, this will include your full peer review and any attached files.

**Do you want your identity to be public for this peer review?** For information about this choice, including consent withdrawal, please see our Privacy Policy.

Reviewer #1: No

Reviewer #2: No

---

## [Decision Letter · Decision Letter 1]

8 Oct 2024

Cost Effectiveness and Return on Investment Analysis for Surgical Care in a Conflict-Affected Region of Sudan

PGPH-D-24-01395R1

Dear Dr. Rice,

We are pleased to inform you that your manuscript 'Cost Effectiveness and Return on Investment Analysis for Surgical Care in a Conflict-Affected Region of Sudan' has been provisionally accepted for publication in PLOS Global Public Health.

Best regards,

Dhananjaya Sharma, MS, PhD, DSc, FRCS

Academic Editor

Necessary corrections have been made

Reviewer Comments (if any, and for reference):

Reviewer's Responses to Questions

**Comments to the Author**

1. If the authors have adequately addressed your comments raised in a previous round of review and you feel that this manuscript is now acceptable for publication, you may indicate that here to bypass the “Comments to the Author” section, enter your conflict of interest statement in the “Confidential to Editor” section, and submit your "Accept" recommendation.

Reviewer #3: (No Response)

2. Does this manuscript meet PLOS Global Public Health’s publication criteria? Is the manuscript technically sound, and do the data support the conclusions? The manuscript must describe methodologically and ethically rigorous research with conclusions that are appropriately drawn based on the data presented.

Reviewer #3: Yes

3. Has the statistical analysis been performed appropriately and rigorously?

Reviewer #3: Yes

4. Have the authors made all data underlying the findings in their manuscript fully available (please refer to the Data Availability Statement at the start of the manuscript PDF file)?

Reviewer #3: Yes

5. Is the manuscript presented in an intelligible fashion and written in standard English?

Reviewer #3: Yes

6. Review Comments to the Author

Reviewer #3: I would like to congratulate the authors for completing this important study. The comments of the previous reviewers seem to be addressed and I have no further comments to add.

7. PLOS authors have the option to publish the peer review history of their article (what does this mean?). If published, this will include your full peer review and any attached files.

**Do you want your identity to be public for this peer review?** For information about this choice, including consent withdrawal, please see our Privacy Policy.

Reviewer #3: No
